# Pretreatment Neutrophil-to-Lymphocyte Ratio as a Prognostic Marker for the Outcome of HPV-Positive and HPV-Negative Oropharyngeal Squamous Cell Carcinoma

**DOI:** 10.3390/v15010198

**Published:** 2023-01-10

**Authors:** Marius Meldgaard Justesen, Kathrine Kronberg Jakobsen, Simone Kloch Bendtsen, Martin Garset-Zamani, Christine Mordhorst, Amanda-Louise Fenger Carlander, Anita Birgitte Gothelf, Christian Grønhøj, Christian von Buchwald

**Affiliations:** 1Department of Otorhinolaryngology, Head and Neck Surgery and Audiology, Rigshospitalet, University Hospital of Copenhagen, 2100 Copenhagen, Denmark; 2Department of Oncology, Rigshospitalet, University Hospital of Copenhagen, 2100 Copenhagen, Denmark

**Keywords:** oropharyngeal squamous cell carcinoma, OPSCC, human papillomavirus, HPV, neutrophil-to-lymphocyte ratio, NLR

## Abstract

The incidence of oropharyngeal squamous cell carcinoma (OPSCC) has increased in the past decades due to carcinogenic HPV infection. As this patient group suffers from considerable mortality and treatment morbidity it is important to improve prognostic strategies in OPSCC. Inflammation plays a key role in cancer and the neutrophil-to-lymphocyte ratio (NLR) in blood has been suggested as a prognostic factor for OPSCC. This study aimed to investigate the prognostic impact of NLR on overall survival (OS) and recurrence-free survival (RFS) in a retrospective cohort of 1370 patients. Included patients had pretreatment neutrophil and lymphocyte counts available, as well as a known HPV status. Patients were treated with curative intent according to Danish national guidelines. We stratified patients in groups by NLR < 2, NLR 2–4, or NLR > 4 and analyzed the influence of the NLR tertile on OS and RFS. Kaplan–Meier curves illustrated survival probability in OS and RFS in the general cohort and were stratified by HPV status. We found that an increasing NLR was associated with inferior OS (HR = 1.5 for NLR > 4) and RFS (HR = 1.6 for NLR 2–4; HR = 1.8 for NLR > 4) in multivariable analysis. The Kaplan–Meier curves displayed inferior OS and RFS with an increasing NLR for both HPV+ and HPV− patients. In conclusion, we showed that an increasing NLR is prognostic for a worse outcome of OPSCC independently of HPV status. There are possible uses of NLR in prognostication and treatment de-escalation although further studies are warranted to determine the clinical utility.

## 1. Introduction

Through the past decades, there has been an increase in the incidence of oropharyngeal squamous cell carcinomas (OPSCCs), notably in the Western World [1]. This increase has been driven by an increase in human papillomavirus (HPV)-positive OPSCC [2]. The incidence of oropharyngeal cancer worldwide in 2020 was 98,412 and oropharyngeal cancer resulted in 48,143 new deaths that year [3]. More than 90% of oropharyngeal cancers are classified as squamous cell carcinomas [4]. The 5-year relative survival rate of oropharyngeal cancer has increased from 33.1% in the period 1980–1984 to 58.5% in the period 2010–2014. Despite this improvement in survival, the mortality rate remains more than 40% after five years and it is thus of continued interest to further improve OPSCC treatment strategies and survival [1].

HPV-negative (HPV−) OPSCCs are phenotypically similar to non-oropharyngeal squamous cell carcinomas of the upper aerodigestive tract, with alcohol and tobacco being the main risk factors for the development of malignancy [5]. HPV-positive (HPV+) OPSCCs are most often found in the palatine tonsils or base of tongue, and the carcinogenic relationship between HPV infection and OPSCC has been well established in these anatomical sites [5,6]. HPV status also influences the growth pattern and histological appearance of OPSCC; HPV+ OPSCC tend to develop inside crypts in lingual and palatine tonsils, whereas HPV− OPSCC typically arise from surface epithelium [7].

The HPV status of a tumor has important prognostic value as HPV+ patients generally have a better prognosis than the HPV− OPSCC patients [8]. More than 200 different subtypes of HPV have been identified, and currently 14 subtypes are classified as high-risk for their oncogenic potential [9]. Besides HPV and p16-status, known prognostic factors regarding OPSCC are T stage, N stage, performance status, and smoking history [8].

Inflammation associated with cancer has been suggested to impact the prognosis of malignant disease both regarding overall survival (OS) and disease progression [10]. Increasing neutrophil levels in blood have been shown to correlate with lesser OS in head and neck squamous cell carcinomas (HNSCC) and conversely, increasing lymphocyte levels have been shown to correlate with improved OS [11]. Neutrophil-to-lymphocyte ratio (NLR) is an easily accessible marker for systemic inflammation and has been suggested as a prognostic factor for the outcome of many solid tumor malignancies, including gastrointestinal, hepatocellular, and non–small cell lung carcinomas [10,12]. NLR is a simple ratio of neutrophils and lymphocytes and contains information on the balance between acute and chronic inflammation of innate immunity, represented by neutrophils, and adaptive immunity, represented by lymphocytes [13]. It is of interest to gain more knowledge about easily accessible hematological parameters to further understand the underlying mechanisms in OPSCC as well as to improve the diagnosis and treatment of OPSCC patients.

The aim of this study was to investigate the prognostic impact of NLR on OS and recurrence-free survival (RFS) in a retrospective cohort of 1370 patients diagnosed with OPSCC in Eastern Denmark from 2000 to 2020. Of importance, HPV status was available for all patients and our study was more than double the size of previous studies to include HPV status. The recent increase in the incidence of OPSCC testifies to the significance of investigating factors that are useful in the prognostication of the disease. The large cohort size and general HPV availability in our study makes it ideally suited to contribute with valuable knowledge to the field.

## 2. Materials and Methods

### 2.1. Setting

The study was a retrospective cohort study based on an OPSCC database actively maintained at the Department of Otorhinolaryngology, Head and Neck Surgery and Audiology at Rigshospitalet, Copenhagen University Hospital [14]. Study data in the database were collected and managed using RedCap [15,16].

This cohort included all patients consecutively diagnosed with OPSCC in Eastern Denmark from 2000 to 2020. Currently, the database comprises 2918 patients with OPSCC and is continually updated with new cases. In Denmark, all patients regardless of socioeconomic status are diagnosed and treated in public hospitals and according to The Danish Head and Neck Cancer Study Group’s (DAHANCA) national treatment guidelines, resulting in a non-selected patient cohort [17].

### 2.2. Study Population

HPV+/p16-positive(p16+) tumors were defined as HPV+, whereas HPV+/p16-negative (p16−), HPV−/p16+, or HPV−/p16− tumors were defined as HPV−. Patients with unknown HPV status (*n* = 40) or unknown p16 status (*n* = 43), were excluded from this study. Patients who had distant metastases (*n* = 47) at the time of diagnosis, patients who received palliative therapy (*n* = 133), patients who did not receive any treatment (*n* = 86), or patients who had unknown treatment (*n* = 15) were also excluded. Blood samples were collected prior to treatment initiation and patients without data on neutrophil and lymphocyte counts were excluded (*n* = 1184) (See Appendix A).

The NLR was obtained by dividing the total pretreatment neutrophil count with the total pretreatment lymphocyte count per patient. Ferrandino et al. grouped patients in three groups with NLR-cutoffs for investigating OS of 2.1 and 3.4 while the cutoffs were 2.3 and 3.7 for cancer-specific survival (CSS) [18]. Rachidi et al. examined the effect of NLR on OS in HNSCC and divided patients in three groups with NLR-cutoffs 2.36 and 4.39 [11]. Other studies divided patients in two groups using NLR-cutoffs of 2.42 and 3 [19,20,21]. Based on these previous studies, we chose to separate our patient cohort in three groups of NLR < 2, NLR 2–4, and NLR > 4.

OPSCCs were classified in our database according to UICC 8th. In the novel UICC 8th staging system for OPSCC, different staging criteria are used for p16+ versus p16− carcinomas [22]. The differences in staging between p16+ and p16− carcinomas mainly concern N-staging, although there also is a small difference in T stage, as only p16− carcinomas have a subdivision of T4 in T4a and T4b [23]. In our database, the p16+ carcinomas were classified as either N0, N1, N2, or N3. The p16− carcinomas were classified as N0, N1, N2a, N2b, N2C, N3a, or N3b. We merged the T and N stages of both p16-statuses to be able to treat the study population as a whole entity. p16− N1 and p16+ N1 were grouped as N1. p16− N2a, N2b, and N2c and p16+ N2 were grouped as N2. N3a and N3b of p16− were grouped with N3 of p16+ as N3. T4a and T4b of p16− were grouped with T4 of p16+ as T4. UICC 8th group stages of both p16 statuses were grouped together.

### 2.3. Statistical Methods

R version 4.1.0 (18 May 2021) was used for statistical analyses [24]. Statistical significance was defined by a *p*-value of ≤0.05. A demographic Table 1 stratified by the NLR group was created in R using the package tableone [25]. Packages survminer and survival were used to perform survival analyses and create Kaplan–Meier estimators [26,27].

We performed univariable and multivariable COX proportional hazards regression analysis to investigate the significance of NLR on OS and RFS. The multivariable analysis was adjusted for the variables of age, sex, HPV status, smoking category, performance status, tumor location, T stage, N stage, and NLR tertile. To avoid confounding between treatment type and tumor staging, treatment type was not included in the COX regression analysis, as the clinically appropriate treatment type is chosen depending on stage. Alcohol status was not included in the analyses as a large proportion of patient cases in the study population had unknown alcohol status (*n* = 416, 30.4%) and an inclusion would have hindered our multivariable analysis.

We created Kaplan–Meier survival curves to visualize the survival probability of OS and RFS for the study population. Patients were censored at the last day of follow-up if they had not presented with the outcomes of death or recurrence.

A proportional hazard test with Schoenfeld residuals was used for testing the assumptions for the COX regression analysis of categorical variables (See Appendix A). We found non-proportionality for all variables except the UICC 8th group stage, which was identified as a time-dependent variable. To account for this, the UICC 8th group stage was included as a stratification factor rather than a predictor in our multivariable analysis, thus allowing the baseline hazard function to differ. We tested for the linearity of the continuous covariate patient age by plotting the Martingale residuals against continuous covariates (See Appendix A).

## 3. Results

We included 1370 OPSCC patients with available NLR and HPV status in this study. Two hundred and eighty patients were categorized as NLR < 2, 695 as NLR 2–4, and 395 had an NLR > 4. Median age at diagnosis increased with the increasing NLR group (*p* < 0.001); in the NLR < 2 group median age was 58 years, in the NLR 2–4 group median age was 61 years, and in the NLR > 4 group median age was 63 years. We observed an inverse relationship between the increasing NLR group and distribution of HPV positivity (*p* = 0.001), with 71.8% of tumors being HPV positive in NLR < 2, 64.6% HPV positive in NLR 2–4, and 57.5% HPV positive in NLR > 4. Higher NLR-group entailed a worse performance score (PS) (*p* < 0.001); 79.6% of patients had PS0 in the NLR < 2 group, 72.5% had PS0 in the NLR 2–4 group, whereas 57.2% of patients had PS0 in the NLR > 4 group. The percentage of patients with a high UICC 8th group stage also increased with an increasing NLR (*p* < 0.001); 24.1% of patients in the NLR > 4 group had a UICC 8th group stage IV tumor at diagnosis compared with 17.7% in group NLR 2–4 and 10.0% in group NLR < 2 (Table 1).

Kaplan–Meier curves were created to illustrate the survival probability for OS and RFS stratified by NLR. A significantly lower survival probability was seen with an increasing NLR for both OS and RFS (See Figure 1). We divided the patient cohort into HPV− and HPV+ patients. The same tendency for correlation between the increasing NLR group and inferior survival probability for OS and RFS was observed both the HPV− and the HPV+ cohort (See Figure 2).

In the multivariable analysis, we observed that high NLR influenced OS with HR = 1.5 (*p* = 0.02) in the highest tertile (NLR < 4) with the lowest tertile (NLR < 2) as a reference (See Table 2). When investigating the effect of NLR on RFS we found a HR of 1.6 (*p* = 0.02) for the middle tertile (NLR 2–4) and 1.8 (*p* = 0.003) for the highest tertile (NLR > 4) of NLR with the lowest tertile (NLR < 2) as a reference (See Table 3).

## 4. Discussion

This population-based cohort study, including 1370 patients with OPSCC from the world’s largest consecutive database of HPV-tested OPSCCs, aimed to investigate the prognostic impact of NLR for OS and RFS. In our multivariable analysis for OS, we found a significant HR = 1.5 (*p* = 0.02) for NLR > 4. When investigating the effect on RFS, we found a HR = 1.6 (*p* = 0.02) for NLR 2–4, and HR = 1.8 (*p* = 0.003) for NLR > 4. The Kaplan–Meier survival probability curves illustrated a statistically significant (*p* < 0.0001) difference between NLR groups and OS and RFS with inferior survival rates in the increasing NLR group. This difference was observed both in general and when stratified by HPV status.

Several previous studies on smaller or selected cohorts have been published on NLR as a prognostic marker for the outcome of OPSCC, as well as HNSCC located at other anatomical subsites. In the largest study, Ferrandino et al. included 5840 patients with HNSCC; of these, 1952 patients had OPSCC [18]. In their multivariable subgroup analysis of OS in OPSCC they found HR = 1.25 in the middle tertile (2.1–3.4) and HR = 1.95 for the top tertile (>3.4) of NLR. For CSS they found a HR = 1.48 in the top tertile of NLR. This study was based on US Veteran Affairs patients, 99% of who were male, and did not include HPV status. Rachidi et al. included 543 patients with HNSCC of whom 89 patients had HPV-status available [11]. In their multivariable analysis for OS, they found a HR = 1.13 for the middle tertile (2.36–4.39) and HR = 2.39 in the highest tertile (>4.39) of NLR. Ng et al. included 848 patients of whom 674 had HPV status available [20]. They found a HR = 1.64 in the multivariable analysis for OS in patients with NLR > 3 compared to NLR < 3. Other smaller studies found NLR to be prognostic for OS, disease-free survival, as well as local and regional tumor control [19,21,28].

In the present study we enrolled 1370 patients, all with available HPV status. This non-selected cohort included patients from a database comprising all patients diagnosed with OPSCC in Eastern Denmark from 2000 to 2020. It is not clear if patients in earlier studies were treated according to a defined treatment guideline, although all were treated with curative intent [11,18,19,20,21,28]. Our cohort solely comprised patients diagnosed and treated according to the Danish national guidelines created by DAHANCA, with treatment being either by radiotherapy, chemoradiotherapy, surgery, or surgery with adjuvant therapy [17].

Previous attempts to determine a normal level of NLR in healthy subjects have yielded reference intervals of 0.78–3.53 and 0.88–4.0 [29,30]. The large inter-individual variation in NLR complicates the decision related to an optimal cutoff value and has resulted in many different cutoff values being used in previous studies investigating NLR and OPSCC [11,18,19,20,21,28,31]. Due to dissimilarity in cutoff values, comparisons of results from different studies should be made with caution. With this reservation in mind, we deemed it acceptable to compare our findings with those of previous studies as the cutoffs used were very near the cutoffs chosen in our study.

Infection can naturally affect the NLR, and it is a possible limitation of our study that we did not adjust for concurrent infection at time of diagnosis [13]. It was not possible to adjust for infection in our analysis, as our OPSCC database does not contain this information. We have no reason to suspect that there would be an uneven distribution of infections at time of diagnosis, skewing towards patients with a certain outcome. In consequence, infections are not expected to have impacted the association between NLR and outcomes OS and RFS. An obvious limitation of the study is that NLR is not a specific biomarker for cancer and can be influenced by comorbidities [18]. Additionally, it has been proposed that an elevated NLR may be caused by an advanced malignant disease concerning tumor mass, nodal stage, and number of metastases [10]. Both comorbidity and advanced disease can naturally decrease OS. The effect of these factors on RFS is more speculative, but one possible explanation could be that clinicians choose a less aggressive, though still curatively intended, treatment plan for patients with comorbidity or advanced disease. A predictable consideration that could be taken about this study is whether there was an actual correlation between NLR and OS and RFS or if the observed correlation was due to comorbidity, advanced tumor stage and HPV status. We excluded patients with distant metastasis at diagnosis as well as patients who did not receive curative treatment. Additionally, we adjusted for interacting covariates in our multivariable analysis, performance status being the most indicative of patient comorbidity. These measures significantly reduced the possibility of confounding by comorbidity and advanced disease on the correlation found between NLR and RFS or OS.

The tumor microenvironment (TME) plays an important role in the progression of OPSCC [32]. In several solid cancers, thoroughly investigated in melanoma and colorectal cancer, a high density of tumor-infiltrating lymphocytes (TILs) correlates with a better prognosis [33]. This relationship has also been investigated in OPSCC, with a higher infiltration of the tumor with CD4+ and CD8+ T cells associated with a better OS, DSS as well as a lower T stage [32]. This clinical advantage was seen regardless of HPV status, although there was a stronger infiltration in HPV+ tumors. A study by Nordfors et al. found a positive correlation between CD8+ TIL counts and improved OS and DFS in HPV+ OPSCC as well as insignificant trends for effect on OS in HPV− OPSCC. No correlation was found between the infiltration of CD4+ and OS in this study [34]. TIL counts have been suggested as a possible routine prognostic tool in OPSCC [35]. The role of TILs in OPSCC could be, at least in part, an explanation for the improved OS and RFS in patients with lower NLR, although further investigation is needed to expand on whether there is a link between lymphocyte counts, and by extension NLR, and lymphocyte infiltration in tumor tissue.

Current treatment of OPSCC in all treatment modalities is associated with significant morbidity. It is of interest to determine novel strategies for risk stratification and prognostication of OPSCCs for the purpose of treatment de-escalation in patients, where current treatment strategies are unnecessarily aggressive and result in morbidity that could possibly be avoided. Perhaps NLR in combination with HPV/p16 status and TIL density could be included in a such prognostication strategy. Further trials are warranted, especially with focus on confounding comorbidity, for NLR to be included in prognostication for treatment de-escalation.

In conclusion, our Kaplan–Meier survival probability estimators showed reduced OS and RFS in higher groups of NLR. This trend was evident for HPV+ and HPV− separately when stratifying for HPV status. We found inferior RFS with the increasing NLR group, as well as inferior OS in group NLR > 4. Our findings confirmed that a high NLR is a prognostic predictor for inferior OS and RFS, and we found that the significant effect of NLR on OS and RFS was independent of tumor HPV status in the largest non-selected OPSCC cohort including HPV status to date.

## Figures and Tables

**Figure 1 viruses-15-00198-f001:**
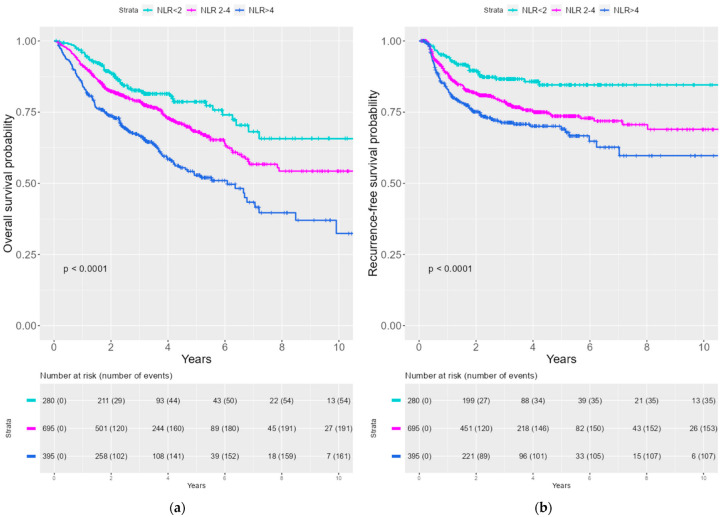
Kaplan–Meier curves of OS and RFS stratified by NLR tertile (**a**) Kaplan–Meier curves of OS by NLR tertile (**b**) Kaplan–Meier curve of RFS by NLR tertile.

**Figure 2 viruses-15-00198-f002:**
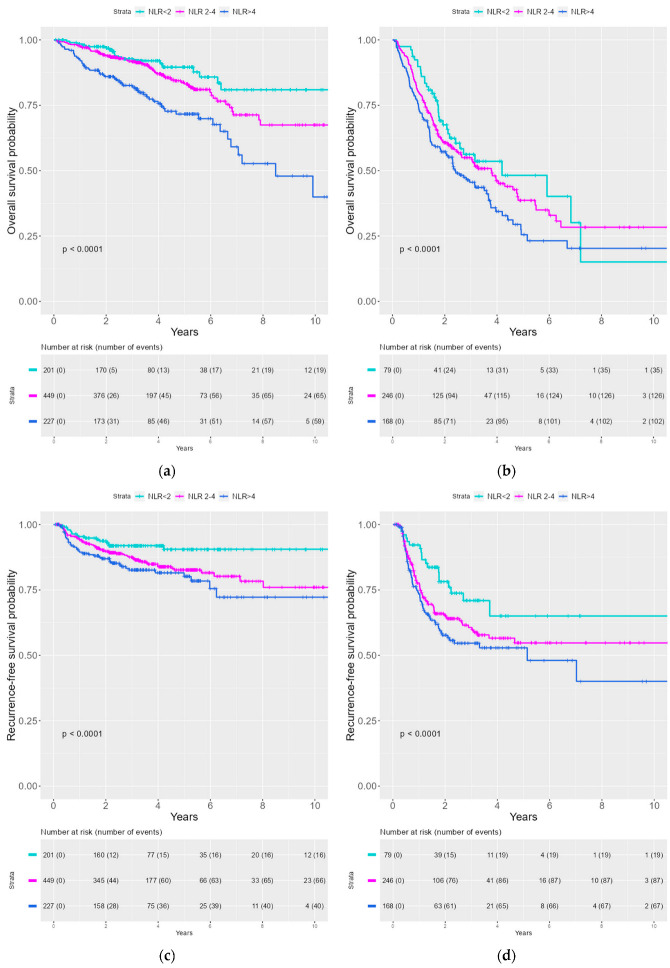
Kaplan–Meier curves of OS and RFS stratified for NLR in HPV+ and HPV− patients. (**a**) OS by NLR for HPV+ patients (**b**) OS by NLR for HPV− patients (**c**) RFS by NLR for HPV+ patients (**d**) RFS by NLR for HPV− patients.

**Table 1 viruses-15-00198-t001:** Characteristics of the 1370 patients in the study population.

Variable	Overall	NLR < 2	NLR 2–4	NLR > 4	*p*-Value
	*n* = 1370	*n* = 280	*n* = 695	*n* = 395	
Age, median (IQR)	61 [55;68]	58 [53;65]	61 [54;69]	63 [57;69]	<0.001
Sex, *n* (%)					0.029
Female	380 (27.7)	95 (33.9)	186 (26.8)	99 (25.1)	
Male	990 (72.3)	185 (66.1)	509 (73.2)	296 (74.9)	
HPV status, *n* (%)					0.001
Negative	493 (36.0)	79 (28.2)	246 (35.4)	168 (42.5)	
Positive	877 (64.0)	201 (71.8)	449 (64.6)	227 (57.5)	
Smoking category, *n* (%)					0.08
Never a smoker	326 (23.8)	87 (31.1)	162 (23.3)	77 (19.5)	
0–10 pack years	101 (7.4)	26 (9.3)	47 (6.8)	28 (7.1)	
10–20 pack year	114 (8.3)	16 (5.7)	68 (9.8)	30 (7.6)	
>20 pack years	690 (50.4)	122 (43.6)	347 (49.9)	221 (55.9)	
Unknown	139 (10.1)	29 (10.4)	71 (10.2)	39 (9.9)	
Performance status, *n* (%)					<0.001
PS0	953 (69.6)	223 (79.6)	504 (72.5)	226 (57.2)	
PS1	228 (16.6)	31 (11.1)	115 (16.5)	82 (20.8)	
PS2	40 (2.9)	6 (2.1)	13 (1.9)	21 (5.3)	
PS3	6 (0.4)	0 (0.0)	2 (0.3)	4 (1.0)	
Unknown	143 (10.4)	20 (7.1)	61 (8.8)	62 (15.7)	
Tumor location, *n* (%)					0.480
Pharyngeal wall or unspecified location	132 (9.6)	23 (8.2)	70 (10.1)	39 (9.9)	
Palatine tonsils	711 (51.9)	156 (55.7)	362 (52.1)	193 (48.9)	
Lingual tonsil/Base of tongue or vallecula epiglottica	429 (31.3)	84 (30.0)	218 (31.4)	127 (32.2)	
Soft palate, Uvula, or palatal arches	98 (7.2)	17 (6.1)	45 (6.5)	36 (9.1)	
Treatment type, *n* (%)					0.028
Radiotherapy	517 (37.7)	96 (34.3)	248 (35.7)	173 (43.8)	
Chemo- and radiotherapy	728 (53.1)	159 (56.8)	391 (56.3)	178 (45.1)	
Primary surgery	104 (7.6)	20 (7.1)	47 (6.8)	37 (9.4)	
Primary surgery with adjuvant radiotherapy	13 (0.9)	3 (1.1)	7 (1.0)	3 (0.8)	
Primary surgery with adjuvant chemo-and radiotherapy	8 (0.6)	2 (0.7)	2 (0.3)	4 (1.0)	
T stage, *n* (%)					0.074
T1	352 (25.7)	80 (28.6)	174 (25.0)	98 (24.8)	
T2	542 (39.6)	117 (41.8)	286 (41.2)	139 (35.2)	
T3	272 (19.9)	53 (18.9)	136 (19.6)	83 (21.0)	
T4a or T4b	201 (14.7)	29 (10.4)	97 (14.0)	75 (19.0)	
Unknown	3 (0.2)	1 (0.4)	2 (0.3)	0 (0.0)	
N stage, *n* (%)					<0.001
N0	300 (21.9)	62 (22.1)	131 (18.8)	107 (27.1)	
N1	703 (51.3)	166 (59.3)	372 (53.5)	165 (41.8)	
N2 (including N2a, N2b, N2c)	306 (22.3)	47 (16.8)	164 (23.6)	95 (24.1)	
N3 (including N3a, N3b)	60 (4.4)	5 (1.8)	28 (4.0)	27 (6.8)	
Unknown	1 (0.1)	0 (0.0)	0 (0.0)	1 (0.3)	
UICC 8th group stage, *n* (%)					<0.001
Stage I	606 (44.2)	148 (52.9)	304 (43.7)	154 (39.0)	
Stage II	283 (20.7)	69 (24.6)	137 (19.7)	77 (19.5)	
Stage III	221 (16.1)	33 (11.8)	124 (17.8)	64 (16.2)	
Stage IV	246 (18.0)	28 (10.0)	123 (17.7)	95 (24.1)	
Unknown	14 (1.0)	2 (0.7)	7 (1.0)	5 (1.3)	

**Table 2 viruses-15-00198-t002:** COX Proportional Hazards Regression Analysis, OS.

Variable	Univariable	*p*-Value	Multivariable	*p*-Value
Age	1.0	<0.001	1.0	<0.001
Sex				
Female	ref		ref	
Male	1.2	0.09	1.3	0.02
HPV status				
Negative	ref		ref	
Positive	0.2	<0.001	0.4	<0.001
Smoking category				
Never a smoker	ref		ref	
0–10 pack years	1.3	0.3	1.2	0.6
10–20 pack year	1.6	0.06	0.9	0.8
>20 pack years	4.5	<0.001	2.1	<0.001
Unknown	3.0	<0.001	1.7	0.02
Performance status				
PS0	ref		ref	
PS1	4.1	<0.001	2.0	<0.001
PS2	8.2	<0.001	3.1	<0.001
PS3	43.8	<0.001	12.3	<0.001
Unknown	3.4	<0.001	2.2	<0.001
Tumor location				
Pharyngeal wall or unspecified location	ref		ref	
Palatine tonsils	0.4	<0.001	0.9	0.3
Lingual tonsil/Base of tongue or vallecula epiglottica	0.4	<0.001	0.7	0.05
Soft palate, Uvula, or palatal arches	0.8	0.3	0.7	0.1
T stage				
T1	ref		ref	
T2	1.7	0.001	1.6	0.007
T3	3.5	<0.001	2.3	<0.001
T4a or T4b	4.8	<0.001	2.4	<0.001
Unknown	2.3	0.4	1.1	0.9
N stage				
N0	ref		ref	
N1	0.6	<0.001	1.1	0.7
N2 (including N2a, N2b, N2c)	2.2	<0.001	2.2	<0.001
N3 (including N3a, N3b)	2.0	0.001	2.7	0.001
Unknown	212.5 *	<0.001	5.4	0.2
UICC 8th group stage				
Stage I	ref			
Stage II	2.2	<0.001	-	-
Stage III	4.1	<0.001	-	-
Stage IV	9.3	<0.001	-	-
Unknown	11.3	<0.001	-	-
NLR tertile				
NLR < 2	ref		ref	
NLR 2–4	1.5	0.01	1.2	0.3
NLR > 4	2.5	<0.001	1.5	0.02

* Only one patient was included in this variable analysis and the HR could not be counted as reliable despite statistical significance.

**Table 3 viruses-15-00198-t003:** COX Proportional Hazards Regression Analysis, RFS.

Variable	Univariable	*p*-Value	Multivariable	*p*-Value
Age	1.0	0.7	1.0	0.05
Sex				
Female	ref		ref	
Male	1.3	0.03	1.4	0.3
HPV status				
Negative	ref		ref	
Positive	0.3	<0.001	0.4	<0.001
Smoking category				
Never a smoker	ref		ref	
0–10 pack years	1.5	0.1	1.4	0.2
10–20 pack year	1.7	0.06	1.1	0.8
>20 pack years	3.1	<0.001	1.7	0.004
Unknown	1.6	0.1	1.3	0.4
Performance status				
PS0	ref		ref	
PS1	2.5	<0.001	1.5	0.01
PS2	4.1	<0.001	2.0	0.01
PS3	2.6	0.99	0.0	0.99
Unknown	2.2	<0.001	1.6	0.007
Tumor location				
Pharyngeal wall or unspecified location	ref		ref	
Palatine tonsils	0.7	0.03	1.3	0.3
Lingual tonsil/Base of tongue or vallecula epiglottica	0.8	0.2	1.3	0.2
Soft palate, Uvula, or palatal arches	1.1	0.8	1.1	0.7
T stage				
T1	ref		ref	
T2	1.7	0.002	1.8	0.003
T3	2.9	<0.001	2.4	<0.001
T4a or T4b	3.7	<0.001	2.3	<0.001
Unknown	3.6	0.2	6.9	0.1
N stage				
N0	ref		ref	
N1	0.9	0.3	1.3	0.2
N2 (including N2a, N2b, N2c)	2.7	<0.001	2.1	0.001
N3 (including N3a, N3b)	2.4	0.001	2.3	0.008
Unknown	NA	NA	NA	NA
UICC 8th group stage				
Stage I	ref			
Stage II	1.6	0.01	-	-
Stage III	2.6	<0.001	-	-
Stage IV	6.1	<0.001	-	-
Unknown	4.0	0.02	-	-
NLR tertile				
NLR < 2	ref		ref	
NLR 2–4	1.9	0.001	1.6	0.02
NLR > 4	2.6	<0.001	1.8	0.003

## Data Availability

Data generated at our central large-scale facility available upon reasonable request.

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
