# Peer review of "Pretreatment Neutrophil-to-Lymphocyte Ratio as a Prognostic Marker for the Outcome of HPV-Positive and HPV-Negative Oropharyngeal Squamous Cell Carcinoma"

_viruses, 2023, doi:10.3390/v15010198_

Round 1
Reviewer 1 Report
The authors performed a study to evaluate the baseline Neutrophil-to-Lymphocyte ratio as a prognostic marker for outcome of HPV-positive and HPV-negative oropharyngeal squamous cell carcinoma (OPSCC). This is a well-described retrospective analysis which was well written. I would like to raise some suggestions to improve the paper.
1. The novelty/importance of the study could be further discussed in the introduction section of the manuscript.
2. How many patients did have concurrent active inflammation (e.g. pneumonia) at the time of diagnosis? Did you include or exclude these patients? You should clarify this because these can confound the association between the NLR and OPSCC.
3. There is no information about what treatment the patients underwent. How many patents underwent surgery? Please provide this information.
4. Are tumor-infiltrating lymphocytes (TIL) correlated with serum lymphocyte counts? Could you provide the evidence for that?
5. You used both neutrophil and neutrophile. You’d better use neutrophil because this is more general.
Author Response
Dear reviewer,
Thank you very much for a thorough and very useful review.
Here is our response to your comments:
- Thank you for this suggestion. We have added a sentence on this topic in the introduction, which besides study size and HPV-status availability, should speak to the importance of the study and help set it apart from previous work.
- A very good suggestion. Unfortunately, we do not have information on concurrent inflammation/infection available in our database and therefore cannot include this variable in our study. There is of course a possible confounding mechanism between inflammation and NLR as inflammation naturally affects hematological parameters. We do not expect that this has influenced our results as we have no reason to suspect that there should be more infection/inflammation at time of diagnosis in the group with worse prognosis vs. groups with better prognosis. We added a paragraph on this limitation to the discussion.
- Thank you. A great idea to help specify the patient population. We have added treatment type in the Table 1 of patient characteristics to clarify which treatment the patients underwent. Our group discussed whether to include treatment type in our analysis and decided on not including treatment type in the COX-analysis to avoid confounding with tumor staging (as appropriate treatment type was chosen depending on the tumor stage).
- Thank you for this very good and relevant suggestion. To our knowledge there is no evidence on this matter yet, and we can only hypothesize on the connection between circulating lymphocytes (and in consequence NLR) and TIL. It would be interesting and prudent to investigate this in future studies.
- Good point! We have corrected this, so only neutrophil is used throughout the article.
Reviewer 2 Report
This is a well-written paper about a topic of interest to journal readers. The study explores the potential prognostic role of NLR after curative treatments of oropharyngeal squamous cell carcinoma in a large cohort of patients, including HPV-negative and HPV-positive patients. Although the results of the present study are not new, the paper is scientifically sound, the methods are adequately used, and the results entirely sustain the conclusions.
Only minor recommendation:
Please consider setting the x axis at 1, 2, 3, … 10 years in the Kaplan Meier curves.
Author Response
Dear reviewer,
Thank you very much for your review and feedback.
Regarding your recommendation:
This is a good point, which will improve the figure. The risk table containing "number at risk (number of events)" which accompanies the Kaplan Meier Curve cannot contain all “number at risk” data if the x-axis is split in 1,2,3, ... 10 years in the size and format required in the publication. Instead, we chose to use data breaks of 2 years in the x-axis, setting the x-axis as 0, 2, 4, 6, 8 and 10 years. This allowed the risk table to match the Kaplan Meier curve while still offering a better overview of the data.